# Species of *Dickeya* and *Pectobacterium* Isolated during an Outbreak of Blackleg and Soft Rot of Potato in Northeastern and North Central United States

**DOI:** 10.3390/microorganisms9081733

**Published:** 2021-08-14

**Authors:** Rebecca D. Curland, Amanda Mainello, Keith L. Perry, Jianjun Hao, Amy O. Charkowski, Carolee T. Bull, Ryan R. McNally, Steven B. Johnson, Noah Rosenzweig, Gary A. Secor, Robert P. Larkin, Beth K. Gugino, Carol A. Ishimaru

**Affiliations:** 1Department of Plant Pathology, University of Minnesota, Saint Paul, MN 55108, USA; curl0013@umn.edu (R.D.C.); rmcnally@western.edu (R.R.M.); 2Department of Plant Pathology and Environmental Microbiology, The Pennsylvania State University, University Park, PA 16802, USA; ammainel@ncsu.edu (A.M.); CaroleeBull@psu.edu (C.T.B.); bmk120@psu.edu (B.K.G.); 3School of Integrative Plant Science Plant Pathology and Plant-Microbe Biology Section, Cornell University, Ithaca, NY 14850, USA; KLP3@cornell.edu; 4School of Food and Agriculture, University of Maine, Orono, ME 04469, USA; 5Department of Agricultural Biology, Colorado State University, Fort Collins, CO 80523, USA; Amy.Charkowski@colostate.edu; 6Cooperative Extension, University of Maine, Orono, ME 04469, USA; stevenj@maine.edu; 7Department of Plant, Soil and Microbial Sciences, Michigan State University, East Lansing, MI 48824, USA; rosenzw4@msu.edu; 8Department of Plant Pathology, North Dakota State University, Fargo, ND 58108, USA; Gary.Secor@ndsu.edu; 9USDA ARS, New England Plant, Soil and Water Research Laboratory, University of Maine, Orono, ME 04469, USA; bob.larkin@usda.gov

**Keywords:** blackleg, plant bacteriology, *Pectobacteriaceae*, phylogeny, *Solanum tuberosum*

## Abstract

An outbreak of bacterial soft rot and blackleg of potato has occurred since 2014 with the epicenter being in the northeastern region of the United States. Multiple species of *Pectobacterium* and *Dickeya* are causal agents, resulting in losses to commercial and seed potato production over the past decade in the Northeastern and North Central United States. To clarify the pathogen present at the outset of the epidemic in 2015 and 2016, a phylogenetic study was made of 121 pectolytic soft rot bacteria isolated from symptomatic potato; also included were 27 type strains of *Dickeya* and *Pectobacterium* species, and 47 historic reference strains. Phylogenetic trees constructed based on multilocus sequence alignments of concatenated *dnaJ, dnaX* and *gyrB* fragments revealed the epidemic isolates to cluster with type strains of *D. chrysanthemi*, *D. dianthicola*, *D. dadantii*, *P. atrosepticum*, *P. brasiliense*, *P. carotovorum*, *P. parmentieri*, *P. polaris*, *P. punjabense*, and *P. versatile*. Genetic diversity within D. dianthicola strains was low, with one sequence type (ST1) identified in 17 of 19 strains. *Pectobacterium parmentieri* was more diverse, with ten sequence types detected among 37 of the 2015–2016 strains. This study can aid in monitoring future shifts in potato soft rot pathogens within the U.S. and inform strategies for disease management.

## 1. Introduction

In 2020, potatoes were, by weight, the fifth most produced food crop globally, behind sugar cane, maize, wheat, and rice [1]. The United States was the fifth top country, producing 41.5 billion pounds of potatoes valued at $3.9 billion in 2020 [1,2]. Pectolytic soft rot diseases cause annual field and storage losses in potato production [3,4,5]. Severe outbreaks of soft rot in 2014 in particular led to yield and crop losses across the Northeastern and North Central U.S. [6].

Pectolytic soft rot bacteria can infect potato at any stage in production, from planting to post-harvest storage. Blackleg disease results from infections in mother tubers that spread through vascular tissue and eventually cause dark greasy lesions in the lower stem. Aerial soft rots can be caused by infections of fleshy above-ground tissues including stems and leaves. In potato production, soft rot usually refers to tuber decay in storage, but is also used in general to describe all forms of disease. Infested potatoes, water supplies, equipment, and storage facilities serve as inoculum sources [7]. Symptom development depends on environmental conditions favorable to host susceptibility and to pathogen growth and virulence [6,7,8,9].

Soft rot diseases of potato are caused by several bacterial taxa. Paine described the first known report of the disease in Great Britain [10]. The causal agent was identified as *Bacillus atrosepticus* and the descriptions of morphological characteristics and disease symptoms are almost exactly as described today. Several soft rot bacteria were later classified in the *Enterobacteriaceae* and named as *Erwinia* spp. Currently, most pectolytic soft rot bacteria affecting potato are classified in the genera *Pectobacterium* and *Dickeya* within the family *Pectobacteriaceae* [3,11]. Numerous species of *Pectobacterium* and *Dickeya* are of concern in potato production. The importance of each species varies. Some, like *Dickeya solani* van der Wolf et al. 2014 sp. nov., which emerged in Europe and caused severe losses for several years, have not yet been reported in the U.S. [12]. Zero tolerance laws and international quarantines were established to limit the spread of such highly virulent pathogens. 

The severe 2014 outbreak of blackleg in Maine brought attention to the significance of soft rot diseases in the U.S. potato industry [13]. In 2016, increased losses due to blackleg were reported relative to 2015. In Maine, economic losses in the seed industry resulted from reduced seed emergence and seed disqualifications [14]. In New York, blackleg and aerial soft rot were found at multiple locations, with Long Island incurring significant yield losses [15].

To address continued concerns about the impacts of soft rot diseases on potato production and to increase awareness of the soft rot pathogens present in the U.S., a multi-state survey was conducted to identify the taxa of *Pectobacterium* and *Dickeya* present in the Northeastern and North Central U.S. during 2015–2016. *Dickeya solani* was not detected in any samples collected during the survey; however, the survey led to multiple state-level new, first reports of bacterial species in the region. *Dickeya dianthicola* [5] is now attributed to the devastating losses seen in Maine, New Jersey, and New York [5,13,15,16]. It has since been isolated in Texas and Hawaii [17,18]. *Dickeya solani* has not been isolated from any of the samples collected in 2015–2016. *Pectobacterium*
*parmentieri* [19] emerged from the surveys as another pectolytic soft rot pathogen associated with recent aerial soft rot outbreaks in Maine, New York, Minnesota, North Dakota, and Michigan [15,19,20,21,22]. Although the species *P. parmentieri* was only recently described, it has been in Wisconsin since at least 2001 and is also present in Hawaii [19,23,24,25]. Other *Pectobacterium* species identified from the survey included *P. atrosepticum* and *P. carotovorum* in Maine, and *P. brasiliense* in Minnesota [15,26].

Since 2018, several additional valid species and amended names have been validly published or proposed for *Dickeya* and *Pectobacterium* [27,28,29,30,31,32,33,34,35,36,37]. To increase awareness of the currently recognized soft rot species present in the U.S., we conducted phylogenetic analyses of type strains of several newly named species of *Dickeya* and *Pectobacterium* and soft rot strains from the 2015–2016 survey. To gain insights on the possibility that some of the newly named species were present in older collections, strains isolated prior to 2014 were evaluated as references. The relatedness among strains of *Dickeya dianthicola* and of *P. parmentieri* isolated from different states was also assessed. Our findings confirm that soft rot diseases in the U.S potato industry are caused by a wide range of *Dickeya* and *Pectobacterium* species.

## 2. Materials and Methods

### 2.1. Bacterial Strains

Soft rot bacteria analyzed in this study included 113 strains isolated in 2015 and 2016 from symptomatic potato tissues collected in production and testing fields in Northeastern and North Central USA. Six of the 113 strains, CIR1009, CIR1011, CIR1058, CIR1182, CIR1183, and CIR1185, were obtained from decaying tubers in Minnesota. Eight strains isolated from pond water were also included, giving a total of 121 strains in the 2015–2016 collection (Table 1). The number of strains varied by state: Florida (1), Hawaii (5), Maine (11), Massachusetts (1), Michigan (2), Minnesota (50), New Jersey (1), New York (20), North Dakota (25), and Pennsylvania (5) (Table 1). Isolation and initial classification and identification of bacteria were conducted as previously described [13,15,22,38]. A total of 26 type strains were included in the study (Table 2). Cultures of type and pathotype strains of nine *Dickeya* spp. and six *Pectobacterium* spp. were obtained from the Belgian Co-Ordinated Collections of Micro-Organisms (BCCM/LMG) (Table 2). DNA sequence data of additional type strains of *Dickeya* spp. and *Pectobacterium* spp. were obtained from GenBank (Table 2). DNA or sequences of an additional 40 reference strains of *Dickeya* spp. and eight strains of *Pectobacterium* spp. were obtained from GenBank and ASAP [39], or provided by A. Charkowski from the A. Kelman collection or R.S. Dickey collection (Table 3).

### 2.2. DNA Extraction and Amplification

DNA extractions were performed using DNeasy Blood and Tissue kit (Qiagen). Three loci were targeted for analysis: *dnaJ*, *dnaX,* and *gyrB* [56]. PCR mixtures contained 5 μL GoTaq master mix (Promega, Madison, WI, USA), 0.5 μL dNTPs, 0.125 μL GoTaq polymerase (Promega), 1 μL each forward and reverse primer (10 μM), 16.875 μL sterile H_2_O, and 0.5 μL template DNA. Thermal cycler programs varied for each locus. For *dnaX*, amplification cycles included an initial denaturation for 3 min at 94 °C, 35 cycles consisting of 1 min at 94 °C, 1 min of annealing at 59 °C, a 2 min extension at 72 °C, and a final extension of 5 min at 72 °C. For *dnaJ*, cycle settings consisted of an initial denaturation of 3 min at 94 °C, 35 cycles of 30 s at 94 °C, 30 s annealing at 55 °C, 1 min extension at 72 °C, and a final extension of 10 min at 72 °C. For *gyrB*, the thermal cycler program included a 4 min initial denaturation at 94 °C, 35 cycles of 1 min at 94 °C, 1 min of annealing at 56 °C, 2 min extension at 72 °C, and a final extension of 10 min at 72 °C. PCR products were visualized on 1.0% TBE agarose gel stained with ethidium bromide and shipped to McLabs (San Francisco, CA, USA) for PCR purification and Sanger sequencing.

### 2.3. Multilocus Sequence Analysis (MLSA)

Nucleotide sequences obtained by direct amplification as described above and from GenBank, ASAP, and collaborators, were aligned, trimmed to a consistent length, and concatenated using CLC Main Workbench (Qiagen, Germantown, MD, USA). Fragment lengths for each gene locus were selected as previously described [56]. Sequence fragments and sequences were concatenated in the following order: *dnaJ* (672 bp), *dnaX* (450 bp), *gyrB* (822 bp for *Dickeya* spp., 711 bp for *Pectobacterium* spp.), with a total concatenated length of 1944 bp for *Dickeya* spp. and 1833 bp for *Pectobacterium* spp. Evolutionary model testing was run in CLC Main Workbench which determined the general time reversible model (GTR+G+T) to be the best fit model for this data set. Phylogenies were inferred via Bayesian analysis using Bayesian Evolutionary Analysis Sampling Trees (BEAST 1.8.2) assuming a strict molecular clock and 10 million generations [80]. Output from BEAST was analyzed in Tracer v1.6.0 [81] and phylogenetic trees were constructed in FigTree v1.4.2 (http://tree.bio.ed.ac.uk/software/figtree, accessed on 11 August 2021).

### 2.4. Multilocus Sequence Typing (MLST)

Sequence types were assigned for 25 strains of *D. dianthicola* and 39 strains of *P. parmentieri* to further characterize diversity within these groups. Unique haplotypes for each locus and for concatenated sequences were identified using DnaSP v5 [82]. Minimum spanning trees were constructed in PHYLOViZ 2.0 to infer distance-based relativity of STs and predict founder STs within clonal complexes [83].

### 2.5. Nucleotide Accession Numbers

Sequences were deposited in GenBank and assigned to following accession numbers: MW978791 to MW979234.

## 3. Results

### 3.1. Phylogeny of 2015–2016 and Reference Strains of Dickeya spp.

The phylogeny predicted by concatenated *dnaJ*, *dnaX*, and *gyrB* sequences placed several reference and 2015–2016 strains, some previously identified only to genus, within clades containing type strains of *D. chrysanthemi*, *D. dianthicola*, *D. dadantii*, and *D. zeae* (Figure 1; Table 1, Table 2 and Table 3). All strains identified previously as *D. dianthicola* were grouped in the same MLSA clade as the *D. dianthicola* type strain, LMG 2485^T^ [13,15,22]. Seven strains in the 2015–2016 collection identified by their providers as *Dickeya* sp. Were also grouped with LMG 2485^T^, as was reference strain *Dickeya* sp. 600, which was isolated from a sweet potato in Georgia. Five potato strains from Hawaii were most closely related to the type strain of *D. dadantii* subsp. *dadantii*, as were four strains from maize. Two strains isolated from pond water adjacent to potato fields, PA24 and ST64, in the 2015–2016 collection were related to *D. dianthicola*, while two others, 16H2-68-A and 16H2-68-B were most closely related to *D. zeae*.

Some reference strains, but none of the 2015–2016 strains, were grouped with type strains of *D. aquatica*, *D. chrysanthemi* pv. *parthenii*, *D. solani*, *D. dadantii* subsp. *dieffenbachiae*, *D. fangzhongdai*, or *D. paradisiaca* (Figure 1 and Table 1 and Table 3). Only *D. aquatica* DW 0440, a reference strain isolated from river water in Finland, was identified as *D. aquatica* [54]. In our analyses, one reference strain, *Dickeya* sp. 699, was identified as *D. dadantii* subsp. *dieffenbachiae*, while the subspecies status of *D. dadantii* strain 655 was ambiguous. Reference strain *Dickeya* sp. K1015 was most closely related to *D. chrysanthemi* pv. *parthenii*. The *D. solani* clade contained all six reference strains initially identified as *D. solani*: MK10, MK16, IFB 0099, GBBC 2040, D s0432-1, and RNS08.23.3.1.A. The MLSA phylogeny placed four reference strains recently classified as *D. fangzhongdai* within the same clade as the type strain of this species DSM 101947^T^ (Figure 1, Table 3). 

### 3.2. Phylogeny of 2015–2016 and Reference Strains of Pectobacterium spp.

The phylogenetic tree of *Pectobacterium* placed several strains from the 2015–2016 and reference collections into well-supported clades containing the type strains of *P. atrosepticum*, *P. brasiliense*, *P. carotovorum*, and *P. parmentieri* (Figure 2). In addition, strains of the 2015–2016 collection were most closely related to type strains of *P. polaris*, *P. punjabense*, and *P. versatile*. Minnesota strains CIR1140, CIR1152 and CIR1152 from potato were grouped with *P. polaris* NIBIO 1006^T^. Five strains (CIR1015, CIR1026, CIR1028, CIR1163, CIR1164), all from potato originating in Minnesota or North Dakota, were most closely related to type strain *P. punjabense* SS95^T^. *P. versatile* was the closest relative of three water strains from Maine (16H2-64-A, 16H2-64-B, 16ME-31) and four potato strains isolated in Maine, Minnesota, and North Dakota. Five strains isolated from potato tubers in Minnesota clustered with type strains of either *P. versatile*, *P. parmentieri*, or *P. carotovorum*. The other tuber isolate, CIR1183 was most closely related to *P. versatile* but was on a separate branch nearly as divergent from *P. versatile* as that of *P. polaris* and *P. versatile.* None of the 2015–2016 strains were found in MLSA clades corresponding to *P. actinidae*, *P. aquatica*, *P. aroidearum*, *P. betavasculorum*, *P. cacticida*, *P. fontis*, *P. odoriferum*, *P. peruviense*, *P. polonicum*, *P. wasabiae*, or *P. zantedeschiae*.

Some of the major phylogenetic clades predicted by concatenated *dnaJ*, *dnaX*, and *gyrB* sequence alignments were further subdivided into well-supported sub-lineages. This was especially evident in *P. brasiliense* and *P. carotovorum*, which contained sub-branching of greater divergence than detected within other *Pectobacterium* species. Two major branches of *P. brasiliense* were evident. One, which was further subdivided into four lineages, contained the type strain, eight strains from the 2015–2016 collection, and reference strains LMG 21370 and LMG 21372. The other was comprised of seven strains collected in 2015 and 2016 from Minnesota or North Dakota. Three groups of *P. carotovorum* were detected. Two strains, 16H2-LB and CIR 1080 were located on a branch that was distinct from the major clade containing *P. carotovorum* LMG 2402^T^ (Figure 2, Table 1).

### 3.3. Diversity within Dickeya dianthicola

Twenty-five *Dickeya dianthicola* isolates were included in this study, with six STs identified (Appendix A). Most (17/19) *D. dianthicola* strains from the 2015–2016 collection were assigned sequence type 1 (ST1). ST1 strains came from Maine, New York, Pennsylvania, Massachusetts, and Florida. None of the strains from Netherlands, France, Belgium, or the United Kingdom were ST1. ST2 was found in only one strain isolated in 2016 from New York. The sweet potato strain, 600, from R. S. Dickey’s collection, previously identified as *Dickeya* sp. and identified here as *D. dianthicola*, shared the same sequence type (ST5) as New Jersey strain 16NJ-12 1, and strains RNS0.4.9 and LMG 2485^T^ from France and from the United Kingdom, respectively (Figure 3 and Appendix A).

### 3.4. Diversity within Pectobacterium parmentieri

Several (37) strains in the 2015–2016 collection clustered with type strain, RNS08.42.1A, of *P. parmentieri*, including strains from Michigan, Minnesota, New York, and North Dakota. Ten sequence types were identified within *P. parmentieri* (Figure 4 and Appendix A). *P. parmentieri* RNS08.42.1A^T^ from France was assigned ST2. ST2 was also detected in Minnesota, New York, and North Dakota. Strains with ST5 originated in Finland, Minnesota, New York, and North Dakota. Multiple sequence types were found in some states. For example, strains from Minnesota and New York separated into five and six sequence types, respectively. While some STs, such as ST2 and ST5, were found in multiple states, ST8 and ST10 appeared only in Minnesota and North Dakota, and ST1, ST3, ST4, ST6, ST7, and ST9 were unique to particular states.

## 4. Discussion

The taxonomy of *Pectobacterium* and *Dickeya* has been refined substantially in the past ten years by the widespread adaption of DNA sequencing strategies in soft rot research. Classification and identification of soft rot bacteria by multilocus sequence alignments of concatenated sequences and by whole genome sequence comparisons has added clarification to the complex polyphyletic nature of some earlier-described species of *Pectobacterium* and *Dickeya*. In this study, all strains of *Pectobacterium* and *Dickeya* previously identified to species level by whole genome sequencing grouped with the corresponding type strains in phylogeny predicted by MLSA of *dnaJ*, *dnaX*, and *gyrB*. The clades and phylogenies predicted in this study using concatenated sequences of *dnaJ*, *dnaX*, and *gyrB* are also similar to those generated by single and multiple sequence alignments [15,35,37,51,56,58,63,84,85,86]. The consistency among tree topologies generated by single, multiple, or entire genome sequences enables presumptive identification of unknown soft rot bacteria to species and provides insights on the diversity, ecology, and epidemiology of specific taxa [6,54,85,87,88].

The inclusion of several reference and type strains in our MLSA study provided new insights on species diversity of soft rot bacteria prior to the 2014 blackleg outbreak. MLSA was sufficient for identification of several reference strains previously identified only to the genus level. Reference strains identified as *Dickeya* sp. in Kelman’s and Dickey’s collections were assigned to MLSA clades *D. chrysanthemi*, *D. dadantii* subsp. *dadantii*, *D. dadantii* subsp. *dieffenbachia*, *D. dianthicola*, and *D. zeae.* The placement of the sweet potato strain *Dickeya* sp. 600 within *D. dianthicola* is evidence that *D. dianthicola* was present in the USA prior to the 2014 blackleg outbreak. The four strains identified as *D. fangzhongdai* in the reference collection all originated outside the U.S. *D. fangzhongdai* has very recently been identified in New York as the cause of soft rot of onion [89]. Historic isolates of *Pectobacterium versatile* have been reported from the U.S. In a retrospective study of soft rot bacteria, Portier et al. identified four historic strains from the USA as *P. versatile*: three from potato isolated in 2001 and one from *Iris* sp. isolated in 1973 [37]. Reference strain *Pectobacterium* sp. Ecc71 was recently reclassified as *P. versatile* [30]. In our study, Ecc71 grouped with other *P. versatile* strains. The taxon includes soft rot strains isolated in Russia from potato and cabbage unofficially named “*Candidatus Pectobacterium maceratum*” [30,90]. 

MLSA and MLST of the 2015–2016 collection provides a broad view of the species of pectolytic soft rot bacteria present in the Northeastern and North Central U.S. immediately following the 2014 blackleg outbreak. The identification of *P. parmentieri*, *P. brasiliense*, *P. carotovorum*, and *P. atrosepticum* is consistent with prior reports of the occurrences of these species in the U.S. [15,20,21,23,24,26]. We did not expect to find nine strains of *P. versatile* in the 2015–2016 collection. *P. versatile* was recently reported in potato from New York and appears to represent a shift in 2017 from the *P. parmentieri* that predominated in 2016 [91]. Less expected was finding *P. polaris* and *P. punjabense*, as these have not been reported in the U.S. *P. polaris* was validly reported in 2017 and is found in Norway [52] and Morocco [85]. *P. punjabense* was reported in Pakistan [36]. No strains of *P. polaris* or *P. punjabense* originating from the U.S. were noted in the original species description or in recent examinations of other species of *Pectobacterium* [30,31,35,37,52,84]. Further polyphasic analyses are warranted to validate the MLSA identification of *P. polaris* or *P. punjabense* within the 2015–2016 collection. 

The 2015–2016 collection contained *D. dianthicola* strains originating from multiple states. *D. dianthicola* has been reported previously in New York, Maine, Michigan, and New Jersey [6,13,15,16,22]. Our findings confirmed that *D. dianthicola* is also present in Florida, Massachusetts, and Pennsylvania. The majority (17/19) of *D. dianthicola* isolates in the 2015–2016 collection from Maine, New York, Pennsylvania, Massachusetts, and Florida are sequence type 1 (ST1). This result might be expected if soft rot outbreaks in the Northeastern U.S. were caused by a single (or limited) introduction of one strain of *D. dianthicola*. However, isolates of ST2 and ST5, from New York and New Jersey, respectively, were also identified within the 2015–2016 collection, which suggests the outbreak was not only clonal. A genetic diversity analysis of 256 isolates of *D. dianthicola* collected over 5 years concluded that the blackleg outbreak in Northeastern U.S. was caused by multiple strains [41]. Ge et al., 2021 identified three genotypes within *D. dianthicola*, the frequencies of which varied by year and by state [41]. Some *D. dianthicola* isolates reported by Ge et al. were also included in our study. Based on overlapping results, ST1 in our study corresponds well to Type I of Ge et al., 2021, while ST 2–6 strains are represented in Type II and III [41].

The 2015–2016 collection is not comprehensive, in that the number of strains from the contributing states varied and no systematic sampling strategy was applied for obtaining isolates over states. The variation in coverage by state resulted in different findings. For example, the deep coverage of strains from Minnesota and North Dakota led to first reports of some species and the likely identification of *P. polaris* and *P. punjabense* in the U.S. The abundance of strains of *D. dianthicola* and *P. parmentieri* in the collection enabled sequence type analyses, from which we can conclude that the genetic diversity within *P. parmentieri* was greater than that of *D. dianthicola* during those years and across states. 

Most strains in the 2015-2016 collection originated from plants. The eight strains from water were enough to show that water can serve as a source of *D. dianthicola*, *D. zeae*, *P. actinidae*, and *P. versatile*. In a large comprehensive study of *Dickeya* in temperate regions of Central Europe, the diversity of *Dickeya* species recovered from water was different than that obtained from potato [45]. In that study, *D. dianthicola* was not recovered from water. Future studies involving larger numbers of aquatic strains could improve our understanding of the relative diversity of soft rot species in water and plant sources in the U.S. [6,13,15,16,22,41].

Four newly described species of *Dickeya* were validly published as we completed our analyses [27,29,86,92]. Future studies with two of these, *D. oryzae* and *D. poaceiphila,* would aid in resolution of isolates within the MLSA clade *D. zeae*. Since *D. oryzae* ZYY5^T^ was not included in our study, we did not delineate a clade of *D. oryzae*. NCPPB 3531, CSL RW192, DZ2Q, ZJU 1202, EC1, and the reference strain K1030 from maize clustered together in the MLSA phylogeny presented here. All these strains are currently identified as *D. oryzae* [86]. We also note that two water strains from Maine, 16H2-68-A and 16H2-68-B, and the reference strain, Ech586, are divergent of other strains within the *D. zeae* clade. This might reflect the natural diversity within *D. zeae* or an unresolved species assignment. In a phylogenetic study that included NCPPB 569, the type strain of *D. poaceiphila* [92] and the reference strain Ech586, Ech586 was placed in *D. zeae* [31,93]. It is likely that the water strains are also correctly placed within *D. zeae*; however, because *D. zeae*, *D. poaceiphila*, and *D. chrysanthemi* are closely related species within *Dickeya*, future studies of water strains 16H2-68-A and 16H2-68-B including the type strain of *D. poaceiphila* could verify their species assignment.

*P. parvum* is the only validly published and currently used species name of *Pectobacterium* not included in our study [84]. The inclusion of *P. parvum* might have provided insights on the identity of *Pectobacterium* sp. CIR1183, the 2016 isolate from Minnesota, which by *dnaJ*, *dnaX*, and *gyrB* phylogeny represented a separate branch within the *P. polaris*/*P. versatile* clade. The closest relatives of *P. parvum* are *P. polaris* and *P. versatile*. Isolates of *P. parvum* were described as a group of atypical, less virulent strains closely related to, but distinguished from *P. polaris* [84]. In the future, whole genomic sequence comparisons, evaluation of differentiating biochemical traits, toxicity tests in insects, and soft rot virulence assays that include the type strain of *P. parvum* might enable more accurate placement of *Pectobacterium* sp. CIR1183 within the *P. versatile*/*P. polaris*/*P. parvum* super clade. 

While much progress has been made toward defining a stable, monophyletic taxonomy for species of *Dickeya* and *Pectobacterium*, some taxa remain notably complex and certain strains of soft rot bacteria do not align well with known species [37,63,84]. Two strains, classified here as *P. carotovorum*, CIR1080 and 16H2-LB, clustered together on a separate branch of *P. carotovorum*. These might belong to other species related to *P. carotovorum*. Similarly, the diverging branches of *P. brasiliense* strains within the 2015–2016 collection and reference strains is consistent with the conclusion that *P. brasiliense* is less homogenous than other species of *Pectobacterium* [87].

Phylogenetic relationships predicted by the MLSA schema of *dnaJ*, *dnaX,* and *gyrB* support and extend recent evidence that several newly described species of *Dickeya* and *Pectobacterium* cause soft rot diseases of potato in the U.S. The importance of specific species on disease occurrence, symptomology, and severity remains unclear. MLSA/MLST studies could be extended to study the contributions of within- and between-location genetic variability, multiple species and genotype combinations, and cultivar on disease severity. Comprehensive genomic comparisons of large numbers of representative strains within a given species will continue to advance stable, well-delineated species definitions with the goal of improving pathogen detection and disease management [6,41,87].

## Figures and Tables

**Figure 1 microorganisms-09-01733-f001:**
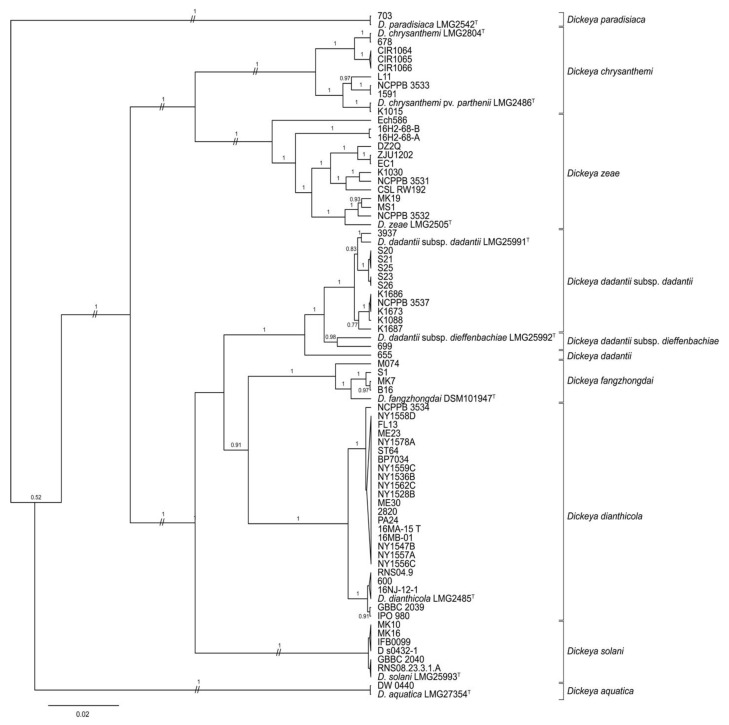
Bayesian tree of *Dickeya* spp. constructed from concatenated sequences of fragments of the genes *dnaJ*, *dnaX*, and gyrB of 29 strains collected in 2015–2016, 10 type strains, and 40 historic reference strains. The superscript letter T indicates the species type strain. Posterior probabilities > 0.6 are shown at the corresponding node. Branch lengths are drawn to scale and represent sequence changes since the common ancestor.

**Figure 2 microorganisms-09-01733-f002:**
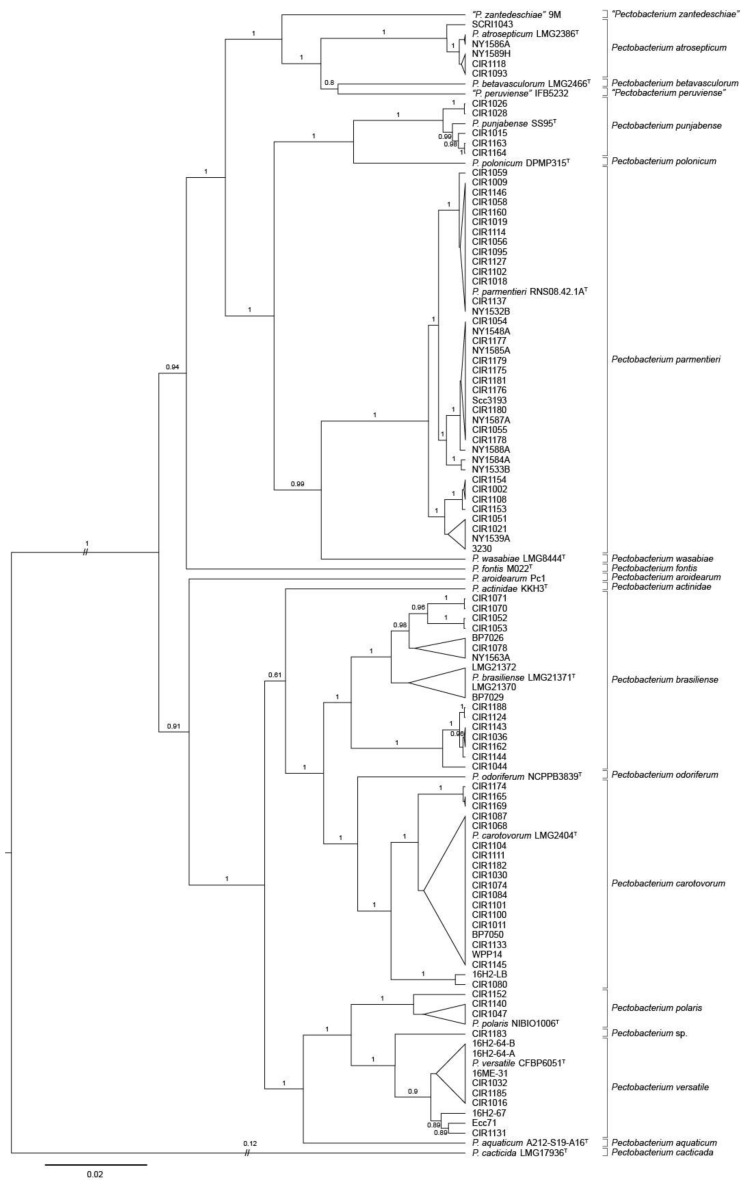
Bayesian tree of *Pectobacterium* spp. constructed from concatenated sequences of fragments of the genes *dnaJ*, *dnaX*, and *gyrB* of 92 strains collected in 2015–2016, 17 type strains, and seven historic reference strains. The superscript letter T indicates the species type strain. Posterior probabilities > 0.6 are shown at the corresponding node. Branch lengths are drawn to scale and represent sequence changes since the common ancestor.

**Figure 3 microorganisms-09-01733-f003:**
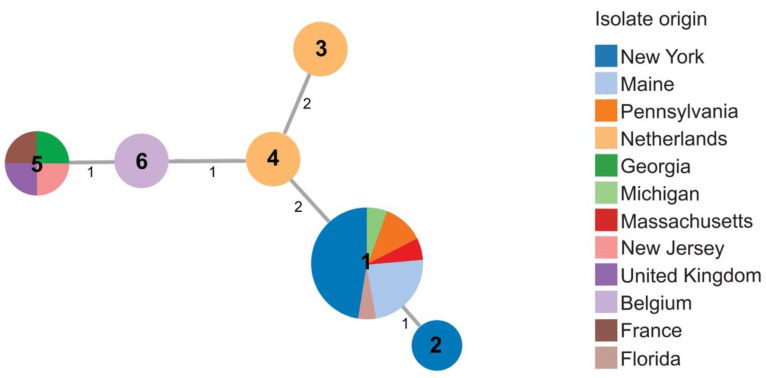
Minimal spanning tree showing relatedness of 25 strains of *D. dianthicola* including 19 strains obtained in 2015–2016. Each circle represents a unique sequence type (ST) derived from the concatenated sequences of three housekeeping genes (*dnaJ*, *dnaX*, and *gyrB*). Sequence type numbers are given in black in each circle. Sizes of circles are relative to the number of individuals sharing the same ST. The relatedness between strains is indicated by relative distance as indicated by branch lengths. Geographic origins of strains are depicted by color: New York (blue); Maine (light blue); Pennsylvania (orange); Netherlands (light orange); Georgia (green); Michigan (light green); Massachusetts (red); New Jersey (pink); United Kingdom (purple); Belgium (light purple); France (brown); Florida (light brown).

**Figure 4 microorganisms-09-01733-f004:**
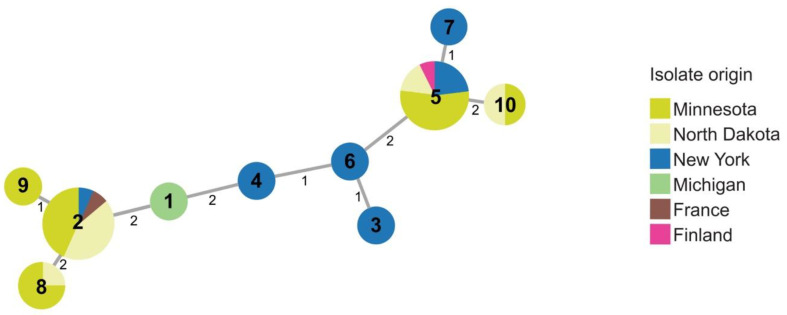
Minimal spanning tree showing relatedness of 39 strains of *P. parmentieri* including 37 strains obtained in 2015–2016. Each circle represents a sequence type based on concatenated STs from three housekeeping genes (*dnaJ*, *dnaX*, and *gyrB*). Sequence type numbers are given in black in each circle. Sizes of circles are relative to the number of individuals sharing the same ST. The relatedness between strains is indicated by relative distance as indicated by branch lengths. Geographic origins of strains are depicted by color: Minnesota (lime green); North Dakota (yellow); New York (blue); Michigan (green); France (brown); Finland (pink).

**Table 1 microorganisms-09-01733-t001:** Description and MLSA clade of *Dickeya* and *Pectobacterium* strains collected in 2015 and 2016 from production and testing areas associated with potato production in Northeastern and North Central U.S.

Initial Identification ^a^	Strain ID	Year Isolated	Geographic Origin	Source Sample	MLSA Clade Identification ^b^	Reference(s)
*Dickeya chrysanthemi*	CIR1064	2016	Minnesota	*Solanum tuberosum*	*Dickeya chrysanthemi*	[38]
*Dickeya dianthicola*	ME23	2015	Maine	*S. tuberosum*	*Dickeya dianthicola*	[15,40]
*Dickeya dianthicola*	ME30	2015	Maine	*S. tuberosum*	*Dickeya dianthicola*	[13]
*Dickeya dianthicola*	2820	2015	Michigan	*S. tuberosum*	*Dickeya dianthicola*	[22]
*Dickeya dianthicola*	PA24	2015	Pennsylvania	water	*Dickeya dianthicola*	[41]
*Dickeya dianthicola.*	16MB-01	2016	Maine	*S. tuberosum*	*Dickeya dianthicola*	[41]
*Dickeya dianthicola*	NY1547B	2016	New York	*S. tuberosum*	*Dickeya dianthicola*	[15]
*Dickeya dianthicola*	NY1556C	2016	New York	*S. tuberosum*	*Dickeya dianthicola*	[15]
*Dickeya dianthicola*	NY1557A	2016	New York	*S. tuberosum*	*Dickeya dianthicola*	[15]
*Dickeya dianthicola*	NY1558D	2016	New York	*S. tuberosum*	*Dickeya dianthicola*	[15]
*Dickeya dianthicola*	NY1559C	2016	New York	*S. tuberosum*	*Dickeya dianthicola*	[15]
*Dickeya dianthicola*	NY1562C	2016	New York	*S. tuberosum*	*Dickeya dianthicola*	[15]
*Dickeya dianthicola*	NY1578A	2016	New York	*S. tuberosum*	*Dickeya dianthicola*	[15]
*Dickeya* sp.	FL13	2016	Florida	*S. tuberosum*	*Dickeya dianthicola*	this study
*Dickeya* sp.	ST64	2016	Maine	water	*Dickeya dianthicola*	this study
*Dickeya* sp.	16MA-15 T	2016	Massachusetts	*S. tuberosum*	*Dickeya dianthicola*	this study
*Dickeya* sp.	16NJ-12 1	2016	New Jersey	*S. tuberosum*	*Dickeya dianthicola*	this study
*Dickeya* sp.	BP7034	2016	Pennsylvania	*S. tuberosum*	*Dickeya dianthicola*	this study
*Dickeya* sp.	16H2-68-A	2016	Maine	water	*Dickeya zeae adjacent*	this study
*Dickeya* sp.	16H2-68-B	2016	Maine	water	*Dickeya zeae adjacent*	this study
*Dickeya* sp.	CIR1065	2016	Minnesota	*S. tuberosum*	*Dickeya chrysanthemi*	this study
*Dickeya* sp.	CIR1066	2016	Minnesota	*S. tuberosum*	*Dickeya chrysanthemi*	this study
*Dickeya* sp.	S20	2015	Hawaii	*S. tuberosum*	*Dickeya dadantii* subsp.* dadantii*	this study
*Dickeya* sp.	S21	2015	Hawaii	*S. tuberosum*	*Dickeya dadantii* subsp.* dadantii*	this study
*Dickeya* sp.	S23	2015	Hawaii	*S. tuberosum*	*Dickeya dadantii* subsp.* dadantii*	this study
*Dickeya* sp.	S25	2015	Hawaii	*S. tuberosum*	*Dickeya dadantii* subsp.* dadantii*	this study
*Dickeya* sp.	S26	2015	Hawaii	*S. tuberosum*	*Dickeya dadantii* subsp.* dadantii*	this study
*Pectobacterium atrosepticum*	NY1586A	2016	New York	*S. tuberosum*	*Pectobacterium atrosepticum*	[15]
*Pectobacterium atrosepticum*	NY1589H	2016	New York	*S. tuberosum*	*Pectobacterium atrosepticum*	[15]
*Pectobacterium brasiliense*	CIR1036 (=SR36)	2015	Minnesota	*S. tuberosum*	*Pectobacterium brasiliense*	[26]
*Pectobacterium brasiliense/Pectobacterium carotovorum*	NY1563A	2016	New York	*S. tuberosum*	*Pectobacterium brasiliense*	[15]
*Pectobacterium brasiliense*	CIR1124 (=SR124)	2016	North Dakota	*S. tuberosum*	*Pectobacterium brasiliense*	[26]
*Pectobacterium brasiliense*	CIR1162 (=SR162)	2016	North Dakota	*S. tuberosum*	*Pectobacterium brasiliense*	[26]
*Pectobacterium carotovorum*	16H2-LB	2016	Maine	water	*Pectobacterium carotovorum*	this study
*Pectobacterium parmentieri*	3230	2015	Michigan	*S. tuberosum*	*Pectobacterium parmentieri*	[22]
*Pectobacterium parmentieri*	CIR1056 (=SR56)	2016	Minnesota	*S. tuberosum*	*Pectobacterium parmentieri*	[21]
*Pectobacterium parmentieri*	NY1532B	2016	New York	*S. tuberosum*	*Pectobacterium parmentieri*	[15]
*Pectobacterium parmentieri*	NY1533B	2016	New York	*S. tuberosum*	*Pectobacterium parmentieri*	[15]
*Pectobacterium parmentieri*	NY1539A	2016	New York	*S. tuberosum*	*Pectobacterium parmentieri*	[15]
*Pectobacterium parmentieri*	NY1548A	2016	New York	*S. tuberosum*	*Pectobacterium parmentieri*	[15]
*Pectobacterium parmentieri*	NY1584A	2016	New York	*S. tuberosum*	*Pectobacterium parmentieri*	[15]
*Pectobacterium parmentieri*	NY1585A	2016	New York	*S. tuberosum*	*Pectobacterium parmentieri*	[15]
*Pectobacterium parmentieri*	NY1587A	2016	New York	*S. tuberosum*	*Pectobacterium parmentieri*	[15]
*Pectobacterium parmentieri*	NY1588A	2016	New York	*S. tuberosum*	*Pectobacterium parmentieri*	[15]
*Pectobacterium* sp.	16ME-31	2016	Maine	*S. tuberosum*	*Pectobacterium versatile*	this study
*Pectobacterium* sp.	CIR1080	2016	Minnesota	*S. tuberosum*	*Pectobacterium carotovorum*	this study
*Pectobacterium* sp.	CIR1118	2016	North Dakota	*S. tuberosum*	*Pectobacterium atrosepticum*	this study
*Pectobacterium* sp.	CIR1093	2016	North Dakota	*S. tuberosum*	*Pectobacterium atrosepticum*	this study
*Pectobacterium* sp.	CIR1044	2015	Minnesota	*S. tuberosum*	*Pectobacterium brasiliense*	this study
*Pectobacterium* sp.	CIR1052	2015	North Dakota	*S. tuberosum*	*Pectobacterium brasiliense*	this study
*Pectobacterium* sp.	CIR1053	2015	North Dakota	*S. tuberosum*	*Pectobacterium brasiliense*	this study
*Pectobacterium* sp.	CIR1143	2016	Minnesota	*S. tuberosum*	*Pectobacterium brasiliense*	this study
*Pectobacterium* sp.	CIR1144	2016	Minnesota	*S. tuberosum*	*Pectobacterium brasiliense*	this study
*Pectobacterium* sp.	CIR1188	2016	Minnesota	*S. tuberosum*	*Pectobacterium brasiliense*	this study
*Pectobacterium* sp.	CIR1070	2016	Minnesota	*S. tuberosum*	*Pectobacterium brasiliense*	this study
*Pectobacterium* sp.	CIR1071	2016	Minnesota	*S. tuberosum*	*Pectobacterium brasiliense*	this study
*Pectobacterium* sp.	CIR1078	2016	Minnesota	*S. tuberosum*	*Pectobacterium brasiliense*	this study
*Pectobacterium* sp.	BP7026	2016	Pennsylvania	*S. tuberosum*	*Pectobacterium brasiliense*	this study
*Pectobacterium* sp.	BP7029	2016	Pennsylvania	*S. tuberosum*	*Pectobacterium brasiliense*	this study
*Pectobacterium* sp.	CIR1011	2015	Minnesota	*S. tuberosum*	*Pectobacterium carotovorum*	this study
*Pectobacterium* sp.	CIR1030	2015	Minnesota	*S. tuberosum*	*Pectobacterium carotovorum*	this study
*Pectobacterium* sp.	CIR1145	2016	Minnesota	*S. tuberosum*	*Pectobacterium carotovorum*	this study
*Pectobacterium* sp.	CIR1165	2016	Minnesota	*S. tuberosum*	*Pectobacterium carotovorum*	this study
*Pectobacterium* sp.	CIR1169	2016	Minnesota	*S. tuberosum*	*Pectobacterium carotovorum*	this study
*Pectobacterium* sp.	CIR1174	2016	Minnesota	*S. tuberosum*	*Pectobacterium carotovorum*	this study
*Pectobacterium* sp.	CIR1182	2016	Minnesota	*S. tuberosum*	*Pectobacterium carotovorum*	this study
*Pectobacterium* sp.	CIR1068	2016	Minnesota	*S. tuberosum*	*Pectobacterium carotovorum*	this study
*Pectobacterium* sp.	CIR1074	2016	Minnesota	*S. tuberosum*	*Pectobacterium carotovorum*	this study
*Pectobacterium* sp.	CIR1084	2016	Minnesota	*S. tuberosum*	*Pectobacterium carotovorum*	this study
*Pectobacterium* sp.	CIR1100	2016	North Dakota	*S. tuberosum*	*Pectobacterium carotovorum*	this study
*Pectobacterium* sp.	CIR1101	2016	North Dakota	*S. tuberosum*	*Pectobacterium carotovorum*	this study
*Pectobacterium* sp.	CIR1104	2016	North Dakota	*S. tuberosum*	*Pectobacterium carotovorum*	this study
*Pectobacterium* sp.	CIR1111	2016	North Dakota	*S. tuberosum*	*Pectobacterium carotovorum*	this study
*Pectobacterium* sp.	CIR1133	2016	North Dakota	*S. tuberosum*	*Pectobacterium carotovorum*	this study
*Pectobacterium* sp.	CIR1087	2016	North Dakota	*S. tuberosum*	*Pectobacterium carotovorum*	this study
*Pectobacterium* sp.	BP7050	2016	Pennsylvania	*S. tuberosum*	*Pectobacterium carotovorum*	this study
*Pectobacterium* sp.	CIR1018	2015	Minnesota	*S. tuberosum*	*Pectobacterium parmentieri*	this study
*Pectobacterium* sp.	CIR1019	2015	Minnesota	*S. tuberosum*	*Pectobacterium parmentieri*	this study
*Pectobacterium* sp.	CIR1002	2015	Minnesota	*S. tuberosum*	*Pectobacterium parmentieri*	this study
*Pectobacterium* sp.	CIR1021	2015	Minnesota	*S. tuberosum*	*Pectobacterium parmentieri*	this study
*Pectobacterium* sp.	CIR1009	2015	Minnesota	*S. tuberosum*	*Pectobacterium parmentieri*	this study
*Pectobacterium* sp.	CIR1051	2015	North Dakota	*S. tuberosum*	*Pectobacterium parmentieri*	this study
*Pectobacterium* sp.	CIR1054	2015	North Dakota	*S. tuberosum*	*Pectobacterium parmentieri*	this study
*Pectobacterium* sp.	CIR1055	2015	North Dakota	*S. tuberosum*	*Pectobacterium parmentieri*	this study
*Pectobacterium* sp.	CIR1146	2016	Minnesota	*S. tuberosum*	*Pectobacterium parmentieri*	this study
*Pectobacterium* sp.	CIR1153	2016	Minnesota	*S. tuberosum*	*Pectobacterium parmentieri*	this study
*Pectobacterium* sp.	CIR1154	2016	Minnesota	*S. tuberosum*	*Pectobacterium parmentieri*	this study
*Pectobacterium* sp.	CIR1175	2016	Minnesota	*S. tuberosum*	*Pectobacterium parmentieri*	this study
*Pectobacterium* sp.	CIR1176	2016	Minnesota	*S. tuberosum*	*Pectobacterium parmentieri*	this study
*Pectobacterium* sp.	CIR1177	2016	Minnesota	*S. tuberosum*	*Pectobacterium parmentieri*	this study
*Pectobacterium* sp.	CIR1178	2016	Minnesota	*S. tuberosum*	*Pectobacterium parmentieri*	this study
*Pectobacterium* sp.	CIR1179	2016	Minnesota	*S. tuberosum*	*Pectobacterium parmentieri*	this study
*Pectobacterium* sp.	CIR1180	2016	Minnesota	*S. tuberosum*	*Pectobacterium parmentieri*	this study
*Pectobacterium* sp.	CIR1181	2016	Minnesota	*S. tuberosum*	*Pectobacterium parmentieri*	this study
*Pectobacterium* sp.	CIR1058	2016	Minnesota	*S. tuberosum*	*Pectobacterium parmentieri*	this study
*Pectobacterium* sp.	CIR1059	2016	Minnesota	*S. tuberosum*	*Pectobacterium parmentieri*	this study
*Pectobacterium* sp.	CIR1102	2016	North Dakota	*S. tuberosum*	*Pectobacterium parmentieri*	this study
*Pectobacterium* sp.	CIR1108	2016	North Dakota	*S. tuberosum*	*Pectobacterium parmentieri*	this study
*Pectobacterium* sp.	CIR1114	2016	North Dakota	*S. tuberosum*	*Pectobacterium parmentieri*	this study
*Pectobacterium* sp.	CIR1127	2016	North Dakota	*S. tuberosum*	*Pectobacterium parmentieri*	this study
*Pectobacterium* sp.	CIR1137	2016	North Dakota	*S. tuberosum*	*Pectobacterium parmentieri*	this study
*Pectobacterium* sp.	CIR1160	2016	North Dakota	*S. tuberosum*	*Pectobacterium parmentieri*	this study
*Pectobacterium* sp.	CIR1095	2016	North Dakota	*S. tuberosum*	*Pectobacterium parmentieri*	this study
*Pectobacterium* sp.	CIR1047	2015	Minnesota	*S. tuberosum*	*Pectobacterium polaris*	this study
*Pectobacterium* sp.	CIR1140	2016	Minnesota	*S. tuberosum*	*Pectobacterium polaris*	this study
*Pectobacterium* sp.	CIR1152	2016	Minnesota	*S. tuberosum*	*Pectobacterium polaris*	this study
*Pectobacterium* sp.	CIR1015	2015	Minnesota	*S. tuberosum*	*Pectobacterium punjabense*	this study
*Pectobacterium* sp.	CIR1026	2015	Minnesota	*S. tuberosum*	*Pectobacterium punjabense*	this study
*Pectobacterium* sp.	CIR1028	2015	Minnesota	*S. tuberosum*	*Pectobacterium punjabense*	this study
*Pectobacterium* sp.	CIR1163	2016	North Dakota	*S. tuberosum*	*Pectobacterium punjabense*	this study
*Pectobacterium* sp.	CIR1164	2016	North Dakota	*S. tuberosum*	*Pectobacterium punjabense*	this study
*Pectobacterium* sp.	CIR1016	2015	Minnesota	*S. tuberosum*	*Pectobacterium versatile*	this study
*Pectobacterium* sp.	CIR1032	2015	Minnesota	*S. tuberosum*	*Pectobacterium versatile*	this study
*Pectobacterium* sp.	16H2-64-A	2016	Maine	water	*Pectobacterium versatile*	this study
*Pectobacterium* sp.	16H2-64-B	2016	Maine	water	*Pectobacterium versatile*	this study
*Pectobacterium* sp.	16H2-67	2016	Maine	water	*Pectobacterium versatile*	this study
*Pectobacterium* sp.	CIR1185	2016	Minnesota	*S. tuberosum*	*Pectobacterium versatile*	this study
*Pectobacterium* sp.	CIR1131	2016	North Dakota	*S. tuberosum*	*Pectobacterium versatile*	this study
*Pectobacterium* sp.	CIR1183	2016	Minnesota	*S. tuberosum*	*Pectobacterium* sp.	this study

^a^ Initial classification of strains from Maine, New York, and Michigan was based on information in cited references. Initial genus classification of strains from Florida, Hawaii, Massachusetts, Minnesota, New Jersey, North Dakota, and Pennsylvania was based minimally on 16S DNA sequence similarities in BLASTN. ^b^ MLSA clades were predicted by Bayesian phylogeny of concatenated alignments of *dnaJ*, *dnaX*, and *gyrB*. Genus and species names correspond to the type strain of the nearest related clade. This study was the source of all amplified and aligned DNA sequences of the 2015–2016 strains. All DNA amplification, sequencing, and sequence editing was performed by the author(s), with the resulting sequences submitted to GenBank (Accession MW978791 to MW979234).

**Table 2 microorganisms-09-01733-t002:** Description of type and pathotype strains used in this study.

Classification ^a^	Type Strain Identifiers ^b^	Year of Isolation	Geographic Origin	Source	Genome Assembly Accession	Source of *dnaJ*, *dnaX*, and *gyrB* ^c^	Reference(s)
*Dickeya aquatica* sp. nov. Parkinson et al., 2014	LMG27354^T^ (=174/2; NCPPB 4580; LMG 27354)	2014	England	river water	GCA_900095885.1	GenBank	[28,42]
*Dickeya chrysanthemi* (Burkholder et al., 1953) Samson et al., 2005, comb. nov.	**LMG 2804^T^** (=ATCC 11663; CCUG 38766; CFBP 2048; CIP 82.99; DSM 4610; ICMP 5703; NCAIM B.01392; NCPPB 402; IPO 2118; Ec17)	1956	USA	*Chrysanthemum morifolium*	GCA_000406105.1	this study	[5,43,44]
*Dickeya chrysanthemi* pv. *parthenii* (Starr 1947) comb. nov.	**LMG 2486^PT^** (=CFBP 1270; ICMP 1547; NCPPB 516; IPO2117)	1957	Denmark	*Parthenium argentatum*	GCA_000406065.1	this study	[5,44]
*Dickeya dadantii* subsp. *dadantii* (Samson et al., 2005) Brady et al., 2012, subsp. nov.	**LMG2 5991^T^** (=Hayward B374; CFBP 1269; ICMP 1544; NCPPB 898)	1960	Comoros	*Pelargonium capitatum*	GCA_000406145.1	this study	[5,44,45,46]
*Dickeya dadantii* subsp. *dieffenbachiae* (Samson et al., 2005) Brady et al., 2012, comb. nov.	**LMG 25992^T^** (=CFBP 2051; ICMP 1568; NCPPB 2976)	1957	USA	*Dieffenbachia* sp.	GCA_000406185.1	this study	[5,44,45,46]
*Dickeya dianthicola* Samson et al., 2005, sp. nov.	**LMG 2485^T^** (=CFBP 1200; ICMP 6427; DSM 18054; NCPPB 453)	1956	United Kingdom	*Dianthus caryophyllus*	GCA_000365305.1	this study	[5,45]
*Dickeya fangzhongdai* Tian et al., 2016, sp. nov.	DSM 101947^T^ (=JS5; CGMCC 1.15464)	2009	China	*Pyrus pyrifolia*	GCA_002812485.1	GenBank	[47]
*Dickeya paradisiaca* (Fernandez-Borrero and Lopez-Duque 1970) Samson et al., 2005, comb. nov.	**LMG 2542^T^** (=ATCC 33242; CFBP 4178; NCPPB 2511)	1970	Columbia	*Musa paradisiaca*	GCA_000400505.1	this study	[5,44,45]
*Dickeya solani* van der Wolf et al., 2014, sp. nov.	**LMG 25993^T^** (=IPO 2222; NCPPB 4479)	2007	Netherlands	*Solanum tuberosum*	GCA_001644705.1	this study	[12,48]
*Dickeya zeae* Samson et al., 2005, sp. nov.	**LMG 2505^T^** (=CFBP 2052; ICMP 5704; NCPPB 2538)	1970	USA	*Zea mays*	GCA_000406165.1	this study	[5,45]
*Pectobacterium actinidae* Portier et al., 2019	KKH3^T^ (=KCTC 23131; LMG 26003)	2012??	Korea	*Actinidia chinensis*	GCA_000803315.1	GenBank	[30,49]
*Pectobacterium aquaticum* sp. nov. Pédron et al., 2019	A212-S19-A16^T^ (=CFBP 8637; NCPPB 4640)	2016	France	water way	GCA-003382565.2	GenBank	[31]
*Pectobacterium atrosepticum* (van Hall 1902) Gardan et al., 2003, comb. nov.	**LMG 2386^T^** (=ATCC 33260; CFBP 1526; CIP 105192; ICMP 1526; NCPPB 549)	1957	United Kingdom	*Solanum tuberosum*	GCA_000749905.1	this study	[4,30]
*Pectobacterium betavasculorum* (Thomson et al., 1984) Gardan et al., 2003, comb. nov.	**LMG 2466^T^** (=ATCC 43762; CFBP 1539; CFBP 2122; CIP 105193; ICMP 4226; LMG 2464; UCPPB 193; NCPPB 2795)	1972	USA	*Beta vulgaris*	GCA_000749845.1	this study	[30,50,51]
*Pectobacterium brasiliense* Portier et al., 2019, sp. nov.	**LMG 21371^T^** (=CFBP 6617; NCPPB 4609)	1999	Brazil	*Solanum tuberosum*	GCA_000754695.1	this study	[30]
*Pectobacterium cacticida* (Alcorn et al., 1991) Hauben et al., 1999, comb. nov.	**LMG 17936^T^** (=1-12; Dye EH-3; ATCC 49481; CFBP 3628; CIP 105191; ICMP 1551-66; ICMP 11136; ICPB EC186; NCPPB 3849)	1958	Arizona	*Carnegiea gigantea*	none available	this study	[30,51]
*Pectobacterium carotovorum* (Jones 1901) Walden 1945 (Approved List 1980)	**LMG 2404^T^** (=ATCC 15713; CFBP 2046; CIP 82.83; DSM 30168; HAMBI 1429; ICMP 5702; NCAIM B.01109; NCPPB 312; VKM B-1247)	1952	Denmark	*Solanum tuberosum*	GCA_900129615.1	this study	[30,51]
*Pectobacterium fontis* sp. nov. Oulghazi et al., 2019	M022^T^ (=CFBP 8629; LMG 30744)	2013	Malaysia	waterfall	GCA_000803215.1	GenBank	[27]
*Pectobacterium odoriferum* (Galloiset al., 1992) Portier et al., 2019 sp. nov.	NCPPB 3839^T^ (=LMG 17566; CFBP 1878; CIP 103762; ICMP 11533)	1978	France	*Cichorium intybus*	GCA_000754765.1	GenBank	[30]
*Pectobacterium parmentieri* Khayi et al., 2016, sp. nov.	RNS 08-42-1A^T^ (=CFBP 8475; LMG 29774)	2008	France	*Solanum tuberosum*	GCA_001742145.1	GenBank	[19,30]
*“Pectobacterium peruviense”* Waleron et al., 2018	IFB 5232^T^ (=PCM 2893; LMG 30269; SCRI 179)	1979	Peru	*Solanum tuberosum*	GCA_002847345.1	GenBank	[33]
*Pectobacterium polaris* Dees et al., 2017, sp. nov.	NIBIO 1006^T^ (=DSM 105255; NCPPB 4611)	2012?	Korea	*Actinidia chinensis*	GCA_002307355.1	GenBank	[52]
*Pectobacterium polonicum* sp. nov. Waleron et al., 2019	DPMP 315^T^ (=PCM 3006; LMG 31077)	2016	Poland	ground water	GCA_005497185.1	GenBank	[35]
*Pectobacterium punjabense* Sarfraz et al., 2018, sp. nov.	SS95^T^ (=CFBP 8604; LMG 30622)	2017	Pakistan	*Solanum tuberosum*	GCA_003028395.1	GenBank	[36]
*Pectobacterium versatile* Portier et al., 2019, sp. nov. Syn. = Candidatus *Pectobacterium maceratum* (Shirshikov et al., 2018)	CFBP 6051^T^ (=NCPPB 3387; ICMP 9168)	pre- 1978	Netherlands	*Solanum tuberosum*	GCA_004296685.1	GenBank	[30,53]
*Pectobacterium wasabiae* (Goto and Matsumoto 1987) Gardan et al., 2003, comb. nov.	**LMG 8404^T^** (=SR91; ATCC 43316; CFBP 3304; CIP 105194; ICMP 9121; NCPPB 3701; PDDCC 9121)	1985	Japan	*Eutrema wasab* *i*	GCA_001742185.1	this study	[4,30]
*“Pectobacterium zantedeschiae”* sp. nov. Waleron et al., 2019	9M^T^ (=PCM 2893; DSM 105717; IFB 9009)	2005	Poland	*Zantedeschia aethiopica*	GCA_004137795.1	GenBank	[34]

^a^ All names of type strains have been validly published, except for those placed within quotations marks. ^b^ Strains in bold type were obtained from Belgian Co-Ordinated Collections of Micro-Organisms (BCCM/LMG) and used to obtain DNA for amplification of *dnaJ*, *dnaX*, and *gyrB* sequences. ^c^ Designates the origin of the data for the sequence fragments of the three loci used in the MLSA phylogenies. “This study” indicates that all DNA amplification, sequencing, and sequence editing was performed by the author(s), with the resulting sequences submitted to GenBank (Accession MW978791 to MW979234). Sequences that were downloaded from genome repositories are indicated as such.

**Table 3 microorganisms-09-01733-t003:** Description of reference strains of *Dickeya* and *Pectobacterium* and MLSA clades predicted by *dnaJ, dnaX*, and *gyrB* multilocus sequence alignments.

Initial Identification ^a^	Strain Identifier(s)	Year of Isolation	Geographic Origin	Sample Origin	MLSA Clade Identification ^b^	GenBank Assembly Accession ^c^	Source of *dnaJ*, *dnaX*, and *gyrB* ^d^	Reference(s)
*Dickeya aquatica*	DW 0440	2005	Finland	river water	*Dickeya aquatica*	GCA_000406285.1	GenBank/ASAP	[28,44,45,54]
*Dickeya chrysanthemi*	NCPPB 3533 (=IPO 655)	1987	USA	*Solanum tuberosum* L.	*Dickeya chrysanthemi*	GCA_000406245.1	GenBank/ASAP	[43,44]
*Dickeya chrysanthemi*	L11	2014	Malaysia	lake water	*Dickeya chrysanthemi*	GCA_000784725.1	GenBank/ASAP	[55]
*Dickeya chrysanthemi*	1591	?	USA (A. Kelman collection)	*Zeae mays*	*Dickeya chrysanthemi*	GCA_000023565.1	GenBank/ASAP	[24,51,56]
*Dickeya dadantii* subsp. *Dadantii*	NCPPB 3537	1987	Peru	*Solanum tuberosum*	*Dickeya dadantii* subsp. *dadantii*	GCA_000406265.1	GenBank/ASAP	[43,44]
*Dickeya dadantii* subsp. *Dadantii*	3937 (=CFBP 3855; Lemattre 3937)	1977	France	*Saintpaulia ionantha*	*Dickeya dadantii* subsp. *dadantii*	GCA_000147055.1	this study	[5,57,58]
*Dickeya dianthicola*	GBBC 2039 (=LMG 25864)	2004	Belgium	*Solanum tuberosum*	*Dickeya dianthicola*	GCA_000365365.1	GenBank/ASAP	[59]
*Dickeya dianthicola*	NCPPB 3534 (=IPO 713)	1987	Netherlands	*Solanum tuberosum*	*Dickeya dianthicola*	GCA_000365405.2	GenBank/ASAP	[59,60].
*Dickeya dianthicola*	IPO 980	?	Netherlands	*Solanum tuberosum*	*Dickeya dianthicola*	GCA_000430955.1	GenBank/ASAP	[58,59]
*Dickeya dianthicola*	RNS04.9	2004	France	*Solanum tuberosum*	*Dickeya dianthicola*	GCA_000975305.1	GenBank/ASAP	[61]
*Dickeya fangzhongdai*	M074	2013	Malaysia	waterfall	*Dickeya fangzhongdai*	GCA_000774065.1	GenBank/ASAP	[62,63]
*Dickeya fangzhongdai*	B16 (=NIB Z 2098)	2010	Slovenia	*Phalaenopsis* sp. (orchid)	*Dickeya fangzhongdai*	GCA_001187965.2	GenBank/ASAP	[62,64]
*Dickeya fangzhongdai*	MK7	?	Scotland	river water	*Dickeya fangzhongdai*	GCA_000406305.1	GenBank/ASAP	[44,62]
*Dickeya fangzhongdai*	S1	2012	Slovenia	*Phalaenopsis* sp.	*Dickeya fangzhongdai*	GCA_001187965.2	GenBank/ASAP	[62,64]
*Dickeya paradisiaca*	Ech703	?	Australia (R. S. Dickey collection)	*Solanum tuberosum*	*Dickeya paradisiaca*	GCA_000023545.1	this study	[24,28,56]
*Dickeya solani*	IFB0099 (=IPO2276)	2005	Poland	*Solanum tuberosum*	*Dickeya solani*	GCA_000831935.2	GenBank/ASAP	[65]
*Dickeya solani*	GBBC 2040	?	Belgium	*Solanum tuberosum*	*Dickeya solani*	GCA_000400565.1	GenBank/ASAP	[59]
*Dickeya solani*	D s0432-1	2004	Finland	*Solanum tuberosum*	*Dickeya solani*	GCA_000474655.1	GenBank/ASAP	[66]
*Dickeya solani*	MK10	?	Israel	*Solanum tuberosum*	*Dickeya solani*	GCA_000365285.1	GenBank/ASAP	[9,59]
*Dickeya solani*	MK16 (=IFB0272; DUC-1)	?	Scotland	river water	*Dickeya solani*	GCA_000365345.1	GenBank/ASAP	[9,45,59]
*Dickeya solani*	RNS08.23.3.1.A	2008	France	*Solanum tuberosum*	*Dickeya solani*	GCA_000511285.2	GenBank/ASAP	[67]
*Dickeya zeae*	NCPPB 3531 (=IPO 645; SCRI 4000)	?	Australia	*Solanum tuberosum*	*Dickeya zeae*	GCA_000406225.1	GenBank/ASAP	[44]
*Dickeya zeae*	NCPPB 3532 (=IPO 646)	?	Australia	*Solanum tuberosum*	*Dickeya zeae*	GCA_000400525.1	GenBank/ASAP	[44]
*Dickeya zeae*	CSL RW192	?	England	river water	*Dickeya zeae*	GCA_000406045.1	GenBank/ASAP	[44]
*Dickeya zeae*	DZ2Q	?	Italy	*Oryza sativa*	*Dickeya zeae*	GCA_000404105.1	GenBank/ASAP	[68]
*Dickeya zeae*	EC1	?	China	*Oryza sativa*	*Dickeya zeae*	GCA_000816045.1	GenBank/ASAP	[69,70]
*Dickeya zeae*	MK19	?	Scotland	river water	*Dickeya zeae*	GCA_000406325.1	GenBank/ASAP	[44]
*Dickeya zeae*	MS1	?	China	*Musa sapientum*	*Dickeya zeae*	GCA_000382585.1	GenBank/ASAP	[71,72]
*Dickeya zeae*	ZJU1202	2012	China	*Oryza sativa*	*Dickeya zeae*	GCA_000264075.1	GenBank/ASAP	[73]
*Dickeya zeae*	Ech586	?	Florida, USA (R. S. Dickey collection)	*Philodendron* sp.	*Dickeya zeae*	GCA_000025065.1	GenBank/ASAP	[24,56]
*Dickeya* sp.	K1015	?	USA (A. Kelman collection)	*Zea mays* convar. *Saccharata* (sweet corn)	*Dickeya chrysanthemi*	n.a.	this study	this study
*Dickeya* sp.	678	?	USA (R. S. Dickey collection)	*Harrisia* (blooming cactus)	*Dickeya chrysanthemi*	n.a.	this study	[24]
*Dickeya* sp.	K1088	?	USA (A. Kelman collection)	*Zeae mays*	*Dickeya dadantii* subsp. *dadantii*	n.a.	this study	this study
*Dickeya* sp.	K1673	?	USA (A. Kelman collection)	*Zeae mays*	*Dickeya dadantii* subsp. *dadantii*	n.a.	this study	this study
*Dickeya* sp.	K1686	?	USA (A. Kelman collection)	*Zeae mays*	*Dickeya dadantii* subsp. *dadantii*	n.a.	this study	this study
*Dickeya* sp.	K1687	?	USA (A. Kelman collection)	*Zeae mays*	*Dickeya dadantii* subsp. *dadantii*	n.a.	this study	this study
*Dickeya* sp.	655	?	Peru (R. S. Dickey collection)	*Ipomoea batatas*	*Dickeya dadantii*	n.a.	this study	this study
*Dickeya* sp.	699	?	Florida, USA (R. S. Dickey collection)	*Alocasia* sp	*Dickeya dadantii* subsp. *dieffenbachiae*	n.a.	this study	this study
*Dickeya* sp.	600	?	Georgia, USA (R. S. Dickey collection)	*Ipomeae batatas*	*Dickeya dianthicola*	n.a.	this study	[24]
*Dickeya* sp.	K1030	?	USA (A. Kelman collection)	*Zeae mays*	*Dickeya zeae*	n.a.	this study	this study
*Pectobacterium aroidearum*	Pc1	2004	Israel	*Ornithogalum dubium*	*Pectobacterium aroidearum*	GCA_000023605.1	GenBank	[74]
*Pectobacterium atrosepticum*	SCRI1043	1985	Scotland	*Solanum tuberosum*	*Pectobacterium atrosepticum*	GCA_000011605.1	GenBank	[75]
*Pectobacterium brasiliense*	LMG 21370 (=CFPB 5507; ATCC BAA-416; Duarte Ecbr 8)	1999	Brazil	*Solanum tuberosum*	*Pectobacterium brasiliense*	n.a.	this study	[30,76]
*Pectobacterium brasiliense*	LMG 21372 (=CFBP 6618; ATCC BAA-418; Duarte Ecbr 213)	1999	Brazil	*Solanum tuberosum*	*Pectobacterium brasiliense*	GCA_000754705.1	this study	[30,63,76]
*Pectobacterium carotovorum*	WPP14	2001	USA	*Solanum tuberosum*	*Pectobacterium carotovorum*	GCA_000173155.1	GenBank	[8,24,77]
*Pectobacterium parmentieri*	Scc3193	1980′s	Finland	*Solanum tuberosum*	*Pectobacterium parmentieri*	GCA_000260925.1	GenBank	[33,78,79]
*Pectobacterium versatile*	Ecc71 (=H.P. Maas Geesteranus/226)	?	Netherlands	*Solanum tuberosum*	*Pectobacterium versatile*	GCA_002983505.1	GenBank	[30,76]

^a^ Identification as given within cited reference or strain collection. ^b^ MLSA clade assignments based on Bayesian analysis of concatenated partial sequences of *dnaJ*, *dnaX*, and *gyrB*. Genus and species names correspond to the type strain of the nearest related clade. ^c^ Designates the origin of the data for the sequence fragments of the three loci used in the MLSA phylogenies. ^d^ “This study” indicates that all DNA amplification, sequencing, and sequence editing was performed by the author(s), with the resulting sequences submitted to GenBank (Accession MW978791 to MW979234). Sequences that were downloaded from genome repositories are indicated as such.

## Data Availability

Not applicable.

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
