# Peer review of "Species of Dickeya and Pectobacterium Isolated during an Outbreak of Blackleg and Soft Rot of Potato in Northeastern and North Central United States"

_microorganisms, 2021, doi:10.3390/microorganisms9081733_

Round 1

Reviewer 1 Report

This study addresses species of soft rot bacteria from potato in northeastern and north central US. It is important to know the diversity and distribution of soft rot bacteria from potato in this huge area. However, a previous study of Ge et al. (2021, Plant Dis.) provided genotyping information of Dickeya dianthicola pathogenic to potato in the US. It needs more discussion between what found in this and provious studies. For instance, what is the relationship between 3 genotypes and sequence types of D. dianthicola.

Moreover, could  you provide the information of potato cultivars in Table 1. Perhaps it has the merit to figure out the relationship between species of soft rot bacteria and potato cultivars.

Minor comments:

  1. The references of bacterial strains listed in Table 1 need correction. For instance, PA24 and 16MB-01 were published in Plant Dis. 2021.
  2. Ln. 43, what is Cwt?
  3. Ln 232, Re 3Some?
  4. Regarding the discussion (Ln 293-301), it was suggested an introduction of the same strain of D. dianthicola caused soft rot outbreaks. It is not clear if seed potatoes planted in Northeastern and Mid-Atlantic US were from the same region.

Reviewer 2 Report

This study analysed the diversity of pectolytic bacteria causing soft rot diseases in US after the severe 2014 outbreak of potato blackleg as well as US reference strains present in older US collections to examine the possible presence of newly described pectolytic bacteria species before the outbreak. It confirms that potato soft rot in US is caused by a wide range of Pectobacteriaceae species. MLST analyses confirm the clonal nature of D. dianthicola strains that spread in US while P. parmentieri strains are more diverse.This is certainly an interesting contribution to the epidemiology of pectolytic bacteria in US.

Besides editing comments (see below), my only concern is the discussion of Clade 1 of Dickeya (lines 305 sqq). First, I don’ understand the sentence “In particular, addition of D. poaceiphila might elucidate the correct assignment of two water strains from Maine, 16H2-68-A and 16H2-68-B, and the reference strain Ech586.” (line 305): Do you imply here that strain Ech586 might belong to D. poaceiphila? This is in contradiction with previous data (Hugouvieux Cotte-Pattat et al, IJSEM 2019; Pédron and Van Gijsegem, OBM genetics 2019) that present ANI data showing the delineation of D. poaceiphila. Please explain more. My second concern is the sentence (line 313): “We speculate all these strains should be considered D. oryzae. »: in fact all these strains (except K1030) were stated to be D. oryzae in the paper describing this species. Please rephrase this §.

Editing comments 

Tables 2 and 3: the 2nd column “strain identifiers” is somewhat difficult to connect to the first column in certain places with a high number of identifiers. Is it possible to add interlines here? Or think about another design?

Line 232: please delete « Re3 »

Reviewer 3 Report

This work analyzed the diversity of 121 SRP strains isolated at the outset of an epidemic in 2015 and 2016 in USA, and also 47 SRP reference strains. The work is correctly conducted but the paper organization is messy and at the end, it is difficult for the reader to find any interest in the results.

Concerning the 121 epidemic strains I am still wondering , if the diversity is different between sampled sites or not ? if some specie are found at a single sampled site?  if strains from particular MLSA sequence types within a given species have been sampled at different location or not ? if there is an evolution between 2015 and 2016 or if you find the same type of pattern between the two years ? if the strains isolated from aerial part and those isolated from tubers belong to different species or not ? Attempts were made to answer some of these questions for D. dianthicola and P. parmentieri but the question remain open for  the other species.

All these analysis are necessary to understand the dynamic of the epidemic and are missing from the present version. Furthermore, the constant mix between the 47 reference strains and the 121 strains isolated during the epidemic in the result and the discussion is obscuring the data.

In conclusion, although the performed work will probably be valuable after an in depth analysis I think the present version should be rejected. Furthermore, MLSA analysis with well-defined reference strains for P. parvum and D. orizae should be performed.

Minor comments

Please, treat separately the 47 reference strains and the 121 epidemic strains both in the result section and in the discussion section.

Concerning the 121 epidemic strains, present your data in a comprehensive way for the reader and analyze your results to clearly answer to the above questions.

Eliminate long taxonomic digression about the species number within Pectobacterium.

Check your references, some are wrongly assigned.

For example you stated

“In a large comprehensive study of Dickeya in temperate regions of Central Europe, water strains were shown to be more diverse than strains from potato”

The reference stated that the diversity is different between the two biomes (potato vs water) not that the diversity is stronger in water

Also

“Pectobacterium versatile is also widely present within French Collection for Plant-Associated Bacteria, CIRM-CFBP, and has been described as pandemic”

The exact reference for this work is not the one you displayed (portier et al, IJSEM 2020) but Portier et al, microorganisms 2020)

Round 2

Reviewer 3 Report

The authors answered all of my concerns, I have got only one small comments

lane 355, please correct "greater" by "different" as diversity observed in water in the cited reference is not greater but different than the one observed on potato

as stated in the summary of potrikus et al

"D. dianthicola and D. zea were isolated from symptomatic potato and ony D. zea and D. chrysanthemi were isolated from water source"

Author Response

we have changed the word as the reviewer suggested. Thank you. The line was 374, not 355.